# Immune Cell Distributions in the Blood of Healthy Individuals at High Genetic Risk of Parkinson’s Disease

**DOI:** 10.3390/ijms252413655

**Published:** 2024-12-20

**Authors:** Laura Deecke, David Goldeck, Olena Ohlei, Jan Homann, Ilja Demuth, Lars Bertram, Graham Pawelec, Christina M. Lill

**Affiliations:** 1Institute of Epidemiology and Social Medicine, University of Münster, Albert-Schweitzer-Campus 1, 48149 Münster, Germany; 2Department of Immunology, University of Tübingen, 72076 Tübingen, Germany; 3Fairfax Centre, Kidlington OX5 2PB, UK; 4Department of Endocrinology and Metabolic Diseases (Including Division of Lipid Metabolism), Charité—Universitätsmedizin Berlin, Corporate Member of Freie Universität Berlin and Humboldt-Universität zu Berlin, Augustenburger Platz 1, 13353 Berlin, Germany; 5BCRT—Berlin Institute of Health Center for Regenerative Therapies, Berlin Institute of Health at Charité—Universitätsmedizin Berlin, 10117 Berlin, Germany; 6Lübeck Interdisciplinary Platform for Genome Analytics (LIGA), University of Lübeck, 23562 Lübeck, Germany; 7Health Sciences North Research Institute, Sudbury, ON P3E 2H3, Canada; 8Ageing Epidemiology Research Unit (AGE), School of Public Health, Imperial College London, London SW7 2AZ, UK

**Keywords:** immune system, neurodegeneration, Parkinson’s disease, polygenic risk score, disease prediction

## Abstract

The immune system likely plays a key role in Parkinson’s disease (PD) pathophysiology. Thus, we investigated whether immune cell compositions are already altered in healthy individuals at high genetic risk for PD. We quantified 92 immune cell subtypes in the blood of 442 individuals using multicolor flow cytometry. Polygenic risk scores (PGS) for PD were calculated based on genome-wide significant SNPs (*n* = 87) from a large genome-wide association study (*n* = 1,530,403). Linear regression analyses did not reveal significant associations between PGS and any immune cell subtype (FDR = 0.05). Nominally significant associations were observed for NKG2C+ B cells (*p* = 0.026) in the overall sample. Older participants at increased genetic PD risk also showed a higher proportion of myeloid dendritic cells (*p* = 0.019) and CD27+CD4+ memory T cells (*p* = 0.043). Several immune cells were nominally statistically associated in women only. These findings suggest that major alterations of immune cells only occur later in the progression of PD.

## 1. Introduction

Idiopathic Parkinson’s disease (PD) is a genetically complex disease determined by a combination and interaction of genetic, environmental, and lifestyle factors. The immune system likely plays a critical role in its development [1]. Along this line, there is increasing evidence for the involvement of different immune cell subtypes in the pathophysiology of PD [2,3]. For example, monocytes are reportedly elevated in PD patients compared to controls, and alterations have also been described for various subsets of T cells [2,3,4]. However, such studies often yield inconsistent results, possibly due to differences in treatment, disease duration, and comorbidities across participants. These discrepancies make it difficult to determine cause–effect relationships, and reverse causation may also be present. Thus, to investigate the earliest changes in immune cell alterations in the course of PD, we explored whether healthy individuals at high genetic risk for PD already show altered immune cell profiles. To this end, we calculated polygenic risk scores (PGS) for PD and analyzed their association with more than 90 immune cell subtypes in the blood of nearly 450 healthy participants from the Berlin Aging Study II (BASE-II; Figure 1).

## 2. Results

The BASE-II dataset analyzed here (*n* = 442, 59% women) consisted of 305 older individuals (59% women, median age: 69 years, range: 60–82 years) and 137 younger individuals (59% women, median age: 29 years, range: 23–35 years, Appendix A).

Linear regression analyses in the BASE-II participants did not reveal any significant associations between the PGS and the 92 immune cell subtypes after FDR control (FDR = 5%), neither in the overall (*n* = 442) nor in the subgroup analyses, i.e., stratifying by sex (i.e., 262 women and 180 men) or analyzing only the older age group (*n* = 305; Appendix A). However, we observed several nominally significant associations: In the overall sample, NKG2C+ B cells tended to be increased in individuals at higher genetic PD risk (variance explained [∆R2] = 1.5%, *p* = 0.026) with consistent effect directions across subgroups and the strongest effect observed in women (∆R2 = 2.9%, *p* = 0.014, Table 1). Furthermore, when restricting our analyses to participants from the old age group, i.e., those closer to a potential onset of PD, higher genetic PD risk was associated with increased proportions of myeloid dendritic cells (∆R2 = 2.5%, *p* = 0.019) and CD27+ CD4+ memory T cells (∆R2 = 1.6%, *p* = 0.043). In addition, women showed inverse associations between PD risk and two immune cell subtypes (myeloid-derived suppressor cells type I [MDSC1], ∆R2 = 2.4%, *p* = 0.024; lineage-negative HLA-DR- cells [Lin-HLA-DR-], ∆R2 = 2.0%, *p* = 0.038), with null effects in men (Table 1). These latter two immune cell types were highly correlated (r = 0.96; Appendix A), likely representing the same association signal, while the other immune cell types showing nominally significant effects in at least one analysis yielded no or only moderate correlations with coefficients between |r| = 0.01 and 0.37. Overall, the total variance explained by the full model for these nominally significant associations ranged from 0.05% to 0.10%, and the variance explained by the PGS ranged from 1.5% to 2.9% (Table 1). The residuals were approximately normally distributed for the vast majority of the linear regression analyses, as determined by visual inspection and the Shapiro–Wilk test for normality (data not shown). This included all analyses yielding nominally significant results (Appendix A). Sensitivity analyses adjusting for CMV status or using the two alternate PGS did not change the results substantially (Appendix A).

## 3. Discussion

To the best of our knowledge, this is the first study to investigate immune cell alterations in healthy individuals at high genetic risk for PD. While no statistically significant changes were detected after FDR control, we observed several nominally significant and functionally relevant alterations, such as increased NKG2C+ B cells (full sample) and higher proportions of myeloid dendritic cells and CD27+ CD4+ memory T cells (old age group). In agreement with the latter observation, CD4+ memory T cells have been reported to be increased in prevalent PD patients compared to controls in previous work [2,6]. In contrast, our observation of increased NKG2C+ B cells with PD risk is novel and warrants further investigation. Notably, B cell subtypes other than NKG2C cells have been reported as either increased [7,8] or decreased [2,9] in PD, though findings across studies remain somewhat inconsistent. Myeloid dendritic cells, which present α-synuclein to T cells and trigger adaptive immune responses in PD, showed a tendency for an increase in PD risk in our study but have also been reported to show a decline with respect to disease severity in diagnosed PD patients [10], indicating a complex role in PD. Interestingly, some immune cell subtypes showed stronger alterations in women than men (NKG2C+ B cells) or were altered only in women but not men (MDSC1, Lin-HLA-DR-), possibly hinting at sex-specific compensatory immune mechanisms, although this remains speculative at this point in time. The variance explained by the PGS remains small (i.e., <3% for any of the top results), which is a finding already described in (endo-)phenotype analyses in other multifactorial traits (e.g., ref. [11,12]).

The main strengths of our study are (i) deep standardized immune cell phenotyping, (ii) comprehensive clinical characterization of the BASE-II participants, and (iii) a comparatively large sample size. However, while we were able to analyze nearly 450 individuals and included more participants than most previous studies on immune cell distributions in PD cases and controls (for review, see [2,3]), power was limited to detect small differences in immune cell composition as may be exerted by polygenic risk scores [12]. In this context and in at least partial agreement with the existing literature, it is possible that the trends of immune cell alterations observed in our study represent genuine findings. Secondly, we investigated individuals at high genetic risk of developing PD, i.e., we did not consider the impact of lifestyle and environmental PD risk factors, and only a fraction of participants may eventually develop PD. The latter may have ‘diluted’ potential prodromal alterations of immune cells in PD pathophysiology. Thirdly, the immune cells analyzed in this study are not entirely independent and exhibit hierarchical relationships. Along these lines, MDSC1 and Lin-HLA-DR- cells, which showed nominally significant associations with the PGS in females, are highly correlated. Consequently, our FDR control may have been rather conservative, potentially increasing the risk of false-negative findings. In this context, it is also conceivable that immune dysfunctions occurring prior to disease are caused by combinatorial effects of multiple immune populations. Future studies with larger sample sizes should consider employing more advanced methods to model the hierarchical structure underlying diverse immune cell subsets. Lastly, we only included participants of European descent, limiting generalizability to other ethnic groups.

In conclusion, immune cell alterations do not seem to be predominant in clinically unaffected participants at high genetic risk of PD. However, our suggestive, nominally significant findings indicating increases of B cell subtypes, myeloid dendritic cells, and memory T cells in individuals at high genetic risk for PD, as well as the potential sex-specific effects, warrant future validation.

## 4. Materials and Methods

### 4.1. Study Participants

This study included healthy adults from BASE-II, a multi-institutional longitudinal study on characteristics of aging [13]. Participants were of European descent, recruited from the Berlin area, and were immunologically (no fever or immune system-related diseases) and cognitively (mini-mental state examination test ≥ 27) healthy with no PD symptoms. Written informed consent was obtained from all participants, and this study was approved by the institutional review boards of all participating institutions. The cohort (*n* = 442) was composed of a group of older individuals and of younger individuals (Appendix A) for whom quality-controlled (QC) genetic as well as immune cell data were available.

### 4.2. Generation and Quality Control of Immune Cell Data by Flow Cytometry

The isolation of peripheral mononuclear blood cells (PBMC) from the whole blood of BASE-II participants and flow cytometry using two different panels was performed as previously described [14,15]: In short, ‘panel 1’ mainly contained T cell subtypes, whereas ‘panel 2’ comprised a range of immune cell subtypes, including natural killer cells, natural killer T cells, monocytes, myeloid-derived suppressor cells (MDSC), B cells, as well as general T cell populations (Appendix A). Antibodies used for the isolation of individual cell types in the panels can be found in Appendix A. Cells were acquired with a 3-laser BD LSRII (BD Biosciences, *Franklin Lakes, NJ, USA*) flow cytometer and DIVA software (https://www.bdbiosciences.com/en-de/products/software/instrument-software/bd-facsdiva-software, accessed on 10 December 2024). Data were analyzed with FlowJo version 7.5 (TreeStar, Woodburn, OR, USA) [14,15]. The immune cells were quantified as blood proportions (with values from 0 to 1). For highly skewed distributions, based on visual inspections, the proportions of immune cells were transformed using log10, root, log(100 − x), and square transformations, respectively. Distributions were subsequently z-transformed (Appendix A). The distributions of the residuals from the linear regression analyses were assessed for normality by visual inspection and using the Shapiro–Wilk test, as implemented in R (https://www.rdocumentation.org/packages/stats/versions/3.6.2/topics/shapiro.test, accessed on 10 December 2024). Notably, the residuals from the linear regression analyses of the untransformed data with skewed distributions frequently showed substantial deviations from normality. Consequently, we restricted our analyses of these data to the transformed versions.

### 4.3. Processing of Genome-Wide SNP Data and Generation of PGS

Genome-wide SNP data were generated using the Affymetrix Array 6.0 and subjected to standard QC as described elsewhere [15]. Ungenotyped genotypes were imputed using the Haplotype Reference Consortium reference [15]. This resulted in 7,512,709 SNPs with a minor allele frequency (MAF) of ≥0.01 and an imputation r^2^ ≥ 0.3. We calculated the PD PGS (Appendix A) based on 87 of the 90 genome-wide, significantly, and independently associated SNPs reported in the largest GWAS on PD risk in Caucasian populations (Table S2 from the original study [5]). To maximize the number of PD risk SNPs included, we relaxed our standard QC thresholds for some variants. This is related to two variants with MAF < 0.01 (i.e., rs114138760 and rs76763715; rs34637584 was monomorphic in our study) and two SNPs showing an imputation r^2^ slightly below 0.3 (r^2^_rs62053943_ = 0.25, r^2^_rs117615688_ = 0.16). Only two SNPs in the human leukocyte antigen (HLA) region were eventually excluded (imputation r^2^ < 0.013). Of note, the 90 independent SNPs described in the original study [5] had been identified using the conditional and joint analysis method (COJO) [16] and included a few SNPs located in the same locus. Thus, in sensitivity analyses, we calculated alternate PGS by (i) only including the most significantly associated SNP per locus (±1 Mb; *n* = 75 SNPs) and (ii) by excluding the four SNPs mentioned above not meeting our stringent QC criteria (Appendix A).

### 4.4. Statistical Analyses

Linear regression analyses of the immune cell distributions on the z-transformed PGS were performed using the lm function in R [17]. The primary analyses were carried out on all study participants. Analyses were adjusted for sex, age group, and the first four genetic principal components (PCs, derived from principal component analyses on genome-wide SNP data generated and processed as previously described [15]) to account for subtle differences in ancestry. The number of principal components was determined using scree plots. Additional subgroup analyses were performed for older participants only (adjusted for sex and genetic PCs) and on the entire dataset after stratifying by sex (adjusted for age group and genetic PCs). Sensitivity analyses included additional adjustments for cytomegalovirus (CMV) status and modifications of the PGS calculation (see above). The false discovery rate (FDR) was controlled at 5%. The correlations among immune cells with at least nominally significant results (*p* < 0.05) were quantified using Spearman’s correlation and visualized in a heatmap.

## Figures and Tables

**Figure 1 ijms-25-13655-f001:**
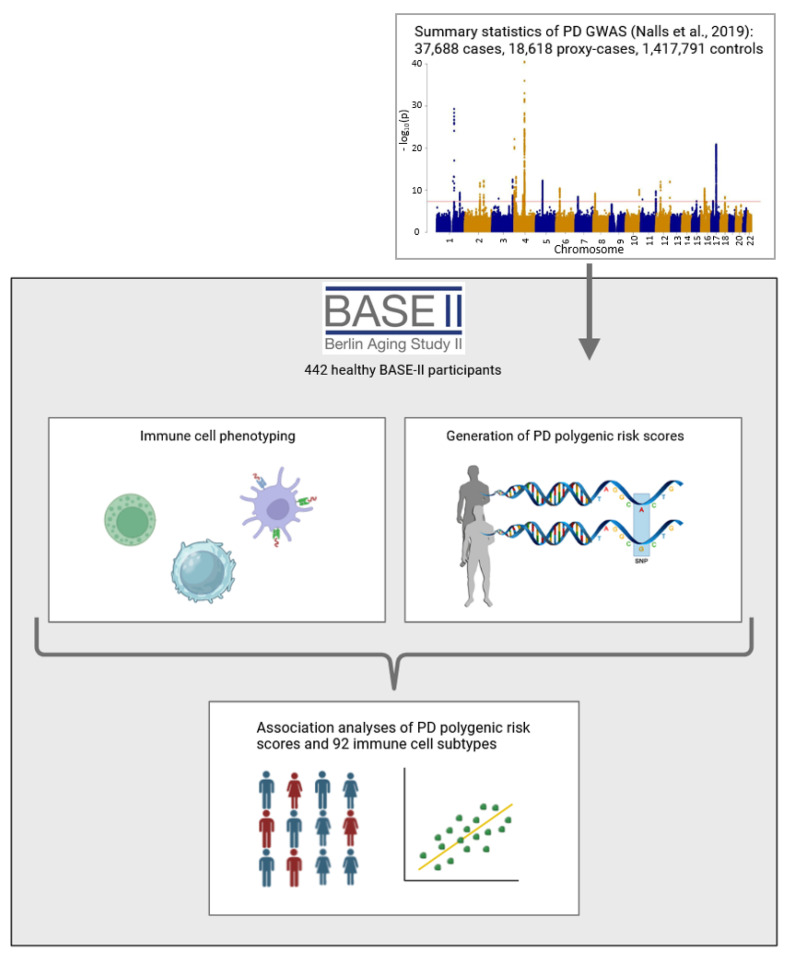
**Study overview.** Schematic overview of the present study generated using BioRender.com. We generated the Manhattan plot of the genome-wide association study based on the summary statistics made publicly available by the authors of the original study [5]. PD = Parkinson’s disease, GWAS = genome-wide association study.

**Table 1 ijms-25-13655-t001:** Association of Parkinson’s disease polygenic risk score with immune cell distributions in the blood of healthy BASE-II participants.

Immune Cell Type	Model	Number	Beta	Full R2	∆R2	*p* Value
NKG2C+ B cells	**All**	320	0.126	0.049	0.015	**0.026**
(CD14-CD3-CD16-CD20+CD3-CD159C+)	Old age	211	0.103	0.054	0.010	0.143
	**Female**	202	0.172	0.078	0.029	**0.014**
	Male	118	0.082	0.109	0.006	0.373
Myeloid dendritic cells	All	320	0.074	0.067	0.005	0.185
(CD14-CD3-CD16+HLA-DR+CD16+)	**Old age**	213	0.163	0.072	0.025	**0.019**
	Female	203	0.047	0.083	0.002	0.496
	Male	117	0.102	0.072	0.010	0.288
CD27+ CD4+ memory T cells	All	375	0.059	0.029	0.003	0.255
(CD3+CD4+CD45RA-CD27+CD28-)	**Old age**	253	0.129	0.051	0.016	**0.043**
	Female	224	−0.008	0.007	0.000	0.905
	Male	149	0.154	0.040	0.023	0.067
Myeloid-derived suppressor cells type I	All	323	−0.088	0.092	0.008	0.108
(MDSC1, Lin-CD14-HLA-DR-CD11b+)	Old age	214	−0.046	0.017	0.002	0.514
	**Female**	205	−0.157	0.084	0.024	**0.024**
	Male	118	0.018	0.181	0.000	0.835
Lineage-negative HLA-DR- cells	All	324	−0.076	0.123	0.006	0.155
(Lin-HLA-DR-, Lin-CD14-HLA-DR-)	Old age	215	−0.031	0.030	0.001	0.655
	**Female**	206	−0.142	0.098	0.020	**0.038**
	Male	118	0.015	0.223	0.000	0.862

Legend. This table displays all nominally significant results (α = 0.05) of the linear regression analyses of immune cell proportions in the blood of healthy participants from the Berlin Aging Study II (BASE-II) on the polygenic score based on genome-wide significant SNPs from the largest available genome-wide association study on Parkinson’s disease in Caucasian populations [5]. Beta = effect estimate of regression analyses of z-transformed immune cell data on z-transformed polygenic risk score. Full R2 = variance explained by the full model (sex, age group, principal components 1 to 4, and PGS as covariates). ∆R2 = variance explained by the polygenic risk score.

## Data Availability

All summary statistics have been made available in the Appendix A of this manuscript. Subject-level data can be obtained by qualified investigators upon request from the authors.

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
