# Peer review of "Immune Cell Distributions in the Blood of Healthy Individuals at High Genetic Risk of Parkinson’s Disease"

_ijms, 2024, doi:10.3390/ijms252413655_

Round 1

Reviewer 1 Report

Comments and Suggestions for Authors

Deckee and colleagues try to find immune cell populations from blood in healthy cohorts that correlate with high genetic risks for Parkinson's Disease. Finding such associations is crucial to identifying novel non-invasive biomarkers for the early diagnosis of PD and will allow treating patients early. While the work is well-motivated and has the potential for great scientific value, I have some reservations about the methodology outlined in the manuscript.

1. "Where applicable, the proportions of immune cells were transformed using log10, root, log(100-x) and square transformations, respectively, and z-transformed." What does "where applicable" mean? What was the initial data before the transformation? Were they defined as proportions from 0 to 1? Also, it would be meaningful to report summary statistics on the total number of cells per condition.

2. The modeling approach lacks some rigor could mask statistically significant associations? Why not analyzing all of the data at once with the appropriate model and instead the data was stratified into 3 cohorts?

3. Somewhat related to 2. In the feature set, there are some hierarchies (e.g., CD4+ T cells in total T cells, CD27+ T cells in total CD4+ T cells). Can the authors justify their choice of not considering this hierarchical relationship and not using, e.g., partial correlation models?

4. What proportion of the PGS variance is explained by the immune cell type distributions? The authors should discuss this aspect in their manuscript.

5. Since no significant associations are found and given the immune system's role in PD, the authors should discuss whether combinatorial effects of immune populations could be better predictors of the disease.

Other minor:

- The modeling methodology needs to be more clear.

- The authors should discuss data availability.

Reviewer 2 Report

Comments and Suggestions for Authors

The authors present a study of individuals at high genetic risk for Parkinson's disease, investigating whether there are alterations in immune cell distributions in these healthy individuals. The authors did not find significant alterations in immune cell distributions, though there were some tendencies towards association in older individuals and in women. 

As acknowledged by the authors, the data and interpretation are limited by several factors, especially sample size, and the lack of inclusion of lifestyle considerations.

In the separate analyses for males and females, did this take into account age? In other words, did the authors separately analyse older males and older females, or was the gender-based analyses done for all ages together?

Overall, while the study is interesting, it is preliminary, without strong data. The authors might consider increasing sample numbers to produce a more robust dataset. 

Round 2

Reviewer 1 Report

Comments and Suggestions for Authors

The authors have sufficiently addressed most of my comments. Given the replies, my only reservation about their analytical approach is the data normalization performed.

Given the authors' approach to normalizing cell rates (Supp Fig 2), my primary concern is whether the normalization of the populations is needed and well justified. In general, linear models do not need to have normally distributed dependent variables, and it looks as if the authors have coerced the data through transformations to be normally distributed. This is unnecessary but acceptable, as long as the transformation does not dilute existing signals. Can the authors show that omitting these transformations and leaving the raw data as is (counts per million as mentioned in Comment 1 reply) affects their conclusions (no significant associations)? For both linear models, the authors should make sure that the residuals follow the normal distribution to ensure that models are applied appropriately. Also, how much of the variation is explained in these models?

Author Response

Comment 1: The authors have sufficiently addressed most of my comments. Given the replies, my only reservation about their analytical approach is the data normalization performed.

Given the authors' approach to normalizing cell rates (Supp Fig 2), my primary concern is whether the normalization of the populations is needed and well justified. In general, linear models do not need to have normally distributed dependent variables, and it looks as if the authors have coerced the data through transformations to be normally distributed. This is unnecessary but acceptable, as long as the transformation does not dilute existing signals. Can the authors show that omitting these transformations and leaving the raw data as is (counts per million as mentioned in Comment 1 reply) affects their conclusions (no significant associations)? For both linear models, the authors should make sure that the residuals follow the normal distribution to ensure that models are applied appropriately.

Response 1: In response to the reviewer’s comment, we have now conducted linear regression analyses on both transformed and untransformed immune phenotypes. As analyzing the proportions of small subsets of specialized cells within total PBMCs does not appear to yield meaningful results we chose to retain the proportions used in our study but without applying log10, root, log(100-x) or square transformations.

Reassuringly, the analyses using non-transformed but skewed data did not yield any significant results that survived FDR correction, and all but one nominally significant findings showed again nominally significant or suggestive (p<0.1) signals (data not shown). However, importantly, examining the residuals revealed an approximately normal distribution for the vast majority of the analyses with transformed data, but substantial deviations from normality for the majority of the analyses with untransformed and skewed data. This observation was statistically supported by the Shapiro-Wilk test. 

In summary, these analyses confirmed that the linear regression models applied to the transformed data represent an appropriate and statistical robust approach for our study. In the revised mansucript, we have now added a dedicated paragraph on the most relevant analyses in this context to each the Methods and Results sections (pages 4 and 7):

The distributions of the residuals from the linear regression analyses were assessed for normality by visual inspection and using the Shapiro-Wilk test, as implemented in R (https://www.rdocumentation.org/packages/stats/versions/3.6.2/topics/shapiro.test). Notably, the residuals from the linear regression analyses of the untransformed data with skewed distributions frequently showed substantial deviations from normality (data not shown). Consequently, we restricted our analyses of these data to the transformed versions.

The residuals were approximately normally distributed for the vast majority of the linear regression analyses, as determined by visual inspection and the Shapiro-Wilk test for normality (data not shown). This included all analyses yielding nominally significant results (Supplementary Figure 4).

We added Supplementary Figure 4 to show the distribution of the residuals for the nominally significant regression analyses. At this point, in order to reduce complexity for the reader, we did not provide the full results of the test statistics or all distribution plots, but would be able to do so if deemed necessary by the reviewer or editor.

Comment 2: Also, how much of the variation is explained in these models?

Response 2: In the original and revised manuscript, we had already provided the incremental r2 contributed by the PGS (“the variance explained for these nominally significant associations ranged from 1.5% to 2.9%”). Following this comment, in the revised manuscript, we now also provide the total variance explained by the model including all covariates (page 7/8, Table 2):

Overall, the total variance explained by the full model for these nominally significant associations ranged from 0.05% to 0.10%, and the variance explained by the PGS ranged from 1.5% to 2.9%.”

Reviewer 2 Report

Comments and Suggestions for Authors

I have no further comments

Author Response

Comment 1: I have no further comments   We thank the reviewer for re-reviewing our manuscript.